# Inflammation, Appetite and Food Intake in Older Hospitalized Patients

**DOI:** 10.3390/nu11091986

**Published:** 2019-08-22

**Authors:** Lars Sieske, Gregor Janssen, Nina Babel, Timm Henning Westhoff, Rainer Wirth, Maryam Pourhassan

**Affiliations:** 1Department of Geriatric Medicine, Marien Hospital Herne, Ruhr-Universität Bochum, 44625 Herne, Germany; 2Medical Department I, General Internal Medicine, Marien Hospital Herne, Ruhr-Universität Bochum, 44625 Herne, Germany

**Keywords:** inflammation, C-reactive protein, malnutrition, appetite, food intake, older persons

## Abstract

The effect of inflammation on appetite and food intake has been rarely studied in humans. In this study, we examined the association of C-reactive protein (CRP), as an inflammatory marker, with appetite and food intake among older hospitalized patients. A total of 200 older individuals, who were consecutively admitted to a geriatric acute care ward, participated in this prospective observational study. Appetite was evaluated using the Edmonton Symptom Assessment System (ESAS) and the Simplified Nutritional Appetite Questionnaire (SNAQ), respectively. Food intake was measured according to plate diagram method and participants were categorized as having food intake <75% and ≥75% of meals served. Nutritional status was evaluated using the Mini Nutritional Assessment Short Form (MNA-SF). In addition, serum CRP was analyzed and the levels >3.0 (mg/dL) were considered as moderate to severe inflammation. Of total population with mean age 81.4 ± 6.6 years (62.5% females), 51 (25.5%) had no inflammation and 88 (44.0%) and 61 (30.5%) had mild and moderate to severe inflammation, respectively. According to MNA-SF, 9.0% and 60.0% had normal nutritional status or a risk of malnutrition, respectively, whereas 31.0% were malnourished. Based on the SNAQ-appetite-question, 32.5% of the patients demonstrated poor and very poor appetite whereas 23.5% reported severe loss of appetite according to ESAS. Ninety-five (48.0%) of the participants had food intake <75% of the meals offered. Significant associations between SNAQ-appetite (*p* = 0.003) and ESAS-appetite (*p* = 0.013) scores and CRP levels were observed. In addition, significant differences were observed in CRP levels between intake ≥75% and <75% of meals served (*p* < 0.001). Furthermore, there were significant associations between appetite and nutritional status whereas malnourished older patients demonstrated a decreased appetite compared to those with normal nutritional status (*p* = 0.011). In a regression analysis, inflammation was the major independent risk factor for patients’ appetite (*p* = 0.003) and food intake (*p* = 0.011) whereas other variables such as infection (*p* = 0.960), chronic inflammatory diseases (*p* = 0.371), age (*p* = 0.679) and gender (*p* = 0.447) do not show any impact on appetite. Our findings confirm that poor appetite and low food intake are associated with inflammation in older hospitalized patients, suggesting that inflammation may contribute an important aspect to the development of malnutrition in these patients.

## 1. Introduction

Malnutrition is a predominant clinical risk factor in older persons, which has detrimental effects on overall health such as physical functioning, quality of life, morbidity and mortality [1]. The process of aging, which is accompanied by physical, physiological and psychological alterations and many concurrent diseases have a negative impact on appetite, and hence, influence food intake [2] and may lead to the development of malnutrition. Up to now, it is unknown which are the most important determinants and causative factors of malnutrition in older persons, although many factors have been identified [3]. However, inflammation is suggested to play an important role in the development of malnutrition, but its association with appetite and food intake has been rarely studied in humans. Indeed, the occurrence of malnutrition, inflammation, and wasting often exist concomitantly in older people, which is referred to the term malnutrition-inflammation-cachexia syndrome (MICS) [4,5]. In addition, Malnutrition Inflammation Score (MIS) is a simple quantitative tool, which determines the existence of nutritional difficulties associated with inflammation [6].

Experimental studies have demonstrated that either peripheral or central infusion of inflammatory mediators may contribute to inhibition of appetite and food intake through interaction in several central nervous system pathways, particularly in the hypothalamus [7,8]. In fact, severe disease associates with tissue inflammation that results in inflammatory responses such as anorexia and fatigue, which may promote negative energy balance and change feeding behavior [9]. In this line, results from a multinational cohort of patients with advanced cancer (age 54–70 years) reported that elevated levels of C-reactive protein (CRP) were significantly associated with appetite loss, cognitive and physical dysfunction, pain and fatigue [10].

CRP has been extensively used as a biomarker of systemic inflammation. Higher levels of CRP were associated with suppressed appetite and food intake in dialysis patients [11,12] and in cancer patients [10]. In addition, the findings of a recent pilot study in geriatric patients similarly confirmed the association between food intake and CRP levels [13]. All together, the findings of these studies suggest that inflammation, measured by elevated CRP may alter involuntary feeding behavior during an acute or chronic inflammatory state, caused by various acute or chronic diseases, including infections. Apart from a few aforementioned studies in humans, which have mainly focused on food intake, not appetite, the relationship between inflammation and appetite has been rarely studied in older hospitalized patients.

Older individuals frequently have inadequate dietary intake and poor appetite, leading to weight loss and increased risk of malnutrition [14,15]. On the other hand, energy demands increase during acute disease and inflammation. Particularly, this mismatch may rapidly lead to disease-related malnutrition [16]. Therefore, reduced food intake and increased energy requirements cause a vicious circle that results in unfavorable health outcomes such as frailty and sarcopenia [17]. It remains a target of interest to clarify to what degree poor appetite and low food intake may be a consequence of inflammation. Therefore, the purpose of the study was to investigate whether there is an association between CRP, as an inflammatory marker, and appetite and food intake among older hospitalized patients. We also aimed to address whether patients’ appetite status reflected food intake.

## 2. Materials and Methods

A total of 200 older participants, who were consecutively admitted to a geriatric acute care ward, participated in this prospective observational study, which was undertaken between September 2017 and November 2018 at the university hospital, Marien Hospital Herne in Germany. Inclusion criteria were patients of 65 years or older who were expected to be hospitalized for at least 7 days, able to cooperate and gave written informed consent. Exclusion criteria were suspected or diagnosed dysphagia, paralysis, severe cognitive impairment (Montreal Cognitive Assessment (MoCA) <10), which all may complicate self-feeding, and artificial nutrition, i.e., tube feeding and parenteral nutrition. Appetite, food intake, inflammation status and geriatric assessment were measured during the first days of hospital admission. The attending physician recorded the clinical routine data. All research related data were obtained and recorded by the first author. The study protocol had been approved by the ethical committee of Ruhr-University Bochum (Nr. 16–5956, approved on 4 April 2017).

### 2.1. Geriatric Assessment

Mini Nutritional Assessment Short Form (MNA-SF) was used to assess nutritional status [18] and subjects were grouped as having normal nutritional status (12–14 points) or a risk of malnutrition (8–11 points) and malnourished (0–7 points). Self-caring activities were determined using Barthel-Index (BI) [19]. The point’s range of the German version of the BI is 0–100 pts., with 100 pts. indicating independence in all activities of daily living. In addition, frailty was evaluated using FRAIL scale [20] with score 0 being not frail, 1–2 pre frail and 3–5 frail. MoCA [21] was used to evaluate cognitive function and patients with total scores below 26 considered as cognitively impaired. Depression in Old Age Scale (DIA-S) [22] was performed to investigate depressive symptoms and subjects were categorized as having no depression (0–2 points), suspected depression (3 point) and probable depression (4–10 points). The risk of sarcopenia was investigated with the use of SARC-F questionnaire [23] that ranges from 0 to 10 and subjects with score ≥4 were defined as having probable sarcopenia. Charlson Comorbidity Index (CCI) [24] was used to determine medical comorbidities.

### 2.2. Assessment of Appetite

Appetite was evaluated using the Edmonton Symptom Assessment System (ESAS) [25] and the Simplified Nutritional Appetite Questionnaire (SNAQ) [26] at time of the hospital admission or the day after. Briefly, the 10-item ESAS comprises nine predefined symptom domains (pain, tiredness, nausea, depression, anxiety, drowsiness, appetite, well-being, shortness of breath) and an optional tenth symptom, which can be defined by the examiner. The numeric analogue scales range from 0 (no symptom) to 10 (worst possible). Accordingly, using ESAS, patients were grouped as having no (0–3 points), average (4–6 points) and severe (7–10 points) loss of appetite.

In addition, patient’s appetite was evaluated using the 4-item SNAQ, which comprises questions about appetite, feeling of fullness, taste of food and the number of meals taken per day. For each question, scaling from 1 (worst possible) to 5, patients indicated the answer that could best apply to their current situation. SNAQ total score is maximum 20, whereas score <14 indicates risk of at least 5% weight loss within six months. Using the respective SNAQ-appetite question, participants were stratified as having very poor and poor (1–2 scores), average (3 scores) and good and very good appetite (4–5 scores).

### 2.3. Assessment of Food Intake

Food intake was determined using the semi-quantitative plate diagram method [27] at time of the hospital admission or the day after. Three senior nurses, who were trained in obtaining plate diagrams due to our clinical routine, were involved in estimation of food intake (intra-class correlation coefficient: 0.91). Briefly, patients’ food intake was estimated for one day by a senior nurse after each main meal and recorded as having eaten all (100%), three-quarters (75%), half (50%), one-quarter (25%) or nothing (0%) of the meal served. Snacks between main meals and nutritional supplements were not considered. In all, final estimation of total food intake was calculated as intake percentage of all main meals divided by three and afterwards participants were categorized as having food intake <75% and ≥75% of meals served.

### 2.4. Measurement of C-Reactive Protein (CRP)

CRP was analyzed according to standard clinical procedures at hospital admission or the day after. Levels between 0.0–0.5 (mg/dL) and between 0.5–3.0 (mg/dL) are considered as no and mild inflammation, respectively. A level >3.0 (mg/dL) is considered as moderate to severe inflammation [28].

### 2.5. Statistical Analysis

The statistical analysis was completed using SPSS statistical software (SPSS Statistics for Windows, IBM Corp, Version 24.0, Armonk, NY, USA). For normally distributed variables, continuous data are reported by means and standard deviations (SDs). Median values are expressed with interquartile ranges (IQR) for non-normally distributed data. Categorical variables are shown as absolute numbers and percentages (*n*, %). Associations of appetite scores and food intake levels with inflammation or nutritional status were tested for significance with Kruskal-Wallis test followed by pairwise comparison or Chi square test as appropriate. In addition, Wilcoxon-Mann Whitney test was performed to assess whether patients’ appetite reflect food intake. Moreover, correlations between SNAQ-appetite and ESAS-appetite scores were calculated using the Spearman’s rank correlation coefficient. An ordinal regression analysis was used to determine the impact of risk factors (i.e., CRP levels, infection, chronic inflammatory diseases, gender and age as independent variables) on patients’ appetite (as dependent variable). A binary logistic regression analysis was also performed to examine the impact of inflammation on food intake while adjusting for other covariates, including frailty, cognitive impairment, depression and nutritional status. *p* < 0.05 was determined as the limit of significance.

## 3. Results:

### 3.1. Subject Characteristics

Baseline characteristics of study participants are summarized in Table 1. Of 200 patients with mean age 81.4 ± 6.6 years (62.5% females), 25.5% had no inflammation and 44.0% and 30.5% had mild and moderate to severe inflammation, respectively. Major reasons of hospitalization were falls and fractures, pneumonia, osteoarthritis, post-stroke care and urinary tract infection. According to MNA-SF, 9.0% and 60.0% had normal nutritional status or a risk of malnutrition, respectively, whereas 31.0% were malnourished. The majority of the subjects were frail (81.0%) and demonstrated an impaired cognitive function (93.0%). In addition, 42.0% of the participants exhibited severe depressive symptoms and 82.0% had probable sarcopenia according to SARC-F. At time of the hospital admission, 15.5% and 11.0% of the patients had infectious (i.e., pneumonia and urinary tract infection) and chronic inflammatory diseases (i.e., rheumatoid arthritis, chronic hepatitis or COPD), respectively. Based on the SNAQ-appetite-question, 32.5% of the patients demonstrated poor and very poor appetite whereas 23.5% reported severe loss of appetite according to ESAS. Ninety-five (48.0%) of the participants had food intake <75% of the meals offered according to plate diagram, in which mean CRP was almost two times higher compared to the patients with food intake ≥75% (4.9 ± 7.3 vs. 2.1± 3.1 mg/dL, respectively; *p* < 0.001).

### 3.2. Association of Inflammation with Appetite and Food Intake

Inflammation and appetite. Significant associations between SNAQ-appetite score and ESAS-appetite score and CRP levels were observed (Table 2). There were significant differences in SNAQ-appetite scores between no and moderate-severe (*p* = 0.010) and between mild and moderate-severe (*p* = 0.009) inflammation but not between no and mild (*p* = 0.732) inflammation (Figure 1). We also observed the similar results using total SNAQ scores across categories of CRP levels with the corresponding significance as *p* = 0.011, *p* = 0.019 and *p* = 0.589, respectively. In addition, the proportions of patients with moderate to severe inflammation became progressively and significantly higher as the SNAQ-appetite scale worsened (from 16.0% in the category good and very good appetite to 44.0% in the category poor and very poor appetite; *p* = 0.011).

There were significant differences in ESAS-appetite score between no and moderate-severe (*p* = 0.010) inflammation but not between mild and moderate-severe (*p* = 0.234) and between no and mild (*p* = 0.406) inflammation. The proportions of patients with moderate to severe inflammation increased as the ESAS-appetite scale worsened, however the difference was not statistically significant (*p* = 0.060).

Furthermore, significant positive associations between ESAS-appetite and nutritional status were observed whereas malnourished older patients demonstrated a decreased appetite compared to those with normal nutritional status (*p* = 0.011). However, such associations were not seen between SNAQ-appetite and nutritional status (*p* = 0.108).

There were significant associations between CRP levels and infection (*p* < 0.001) whereas 61.3% of patients with infection had moderate-severe inflammation. In contrast, appetite scores were not different across infection categories (SNAQ-appetite: *p* = 0.328 and ESAS-appetite: *p* = 0.186) indicating that association between appetite and inflammation is completely independent from infection. Even after excluding patients with acute infection, significant associations between SNAQ-appetite score (*p* = 0.007) and ESAS-appetite score (*p* = 0.016) and CRP levels were observed. In a regression analysis, inflammation (CRP) was the major independent predictor for patients’ appetite (*p* = 0.003) whereas other variables such as infection (*p* = 0.960), chronic inflammatory diseases (*p* = 0.371), age (*p* = 0.679) and gender (*p* = 0.447) did not show any impact on appetite.

Inflammation and food intake. Similar to appetite, inflammation was significantly associated with food intake (Table 2). In addition, 69.0% of patients with moderate-severe inflammation had food intake <75% compared to 30.0% in the category ≥75% of food intake (*p* < 0.001). In a logistic regression analysis, inflammation (*p* = 0.011) and frailty (*p* = 0.023) were the main independent predictors of low food intake whereas other variables such as cognitive impairment (*p* = 0.361), depression (*p* = 0.886) and nutritional status (*p* = 0.499) did not show any effect on low food intake.

### 3.3. Association Between Food Intake and Appetite

Significant associations between food intake and appetite were observed (Table 3). A poor appetite was significantly associated with lower food intake, since almost three-quarters of the total population with very poor and poor SNAQ-appetite (78.0%) or severe loss of appetite (75.6%) according to ESAS had food intake <75% (both *p* < 0.001). In addition, there was a significant negative correlation between SNAQ-appetite and ESAS-appetite scores (*r* = −0.676, *p* < 0.001).

## 4. Discussion

In the present study, we observed positive associations between inflammation and appetite and food intake in older hospitalized patients. Indeed, loss of appetite and low food intake was accompanied by elevated CRP levels. Associations between higher levels of CRP, the most validated inflammatory marker, and poor appetite have been reported in hemodialysis [29,30] and cancer patients [10]. We also have shown in a cross-sectional pilot study that elevated CRP levels were an important influencing factor associated with low food intake in older hospitalized patients [13]. In addition, results of a double-blind nutritional intervention study among hospitalized older patients indicated that participants with increased CRP levels had significantly lower energy intake [31]. However, authors of the latter study analyzed a mixed hospital population and did not assess appetite. To our knowledge, this is the first study investigating the association of inflammation with appetite and food intake in older hospitalized patients.

A major finding of this study was the strong and consistent relationship between loss of appetite and low food intake and higher levels of CRP. We observed that serum concentrations of CRP were almost two-fold higher in patients with food intake <75% of the meals offered. Therefore, our data clearly support previous findings [10,13,30,31] that reported an association of elevated CRP levels with decreased food intake and loss of appetite in other populations. In addition, our study was able to demonstrate that this decrease of nutritional intake is very likely mediated by an inflammation related reduction of appetite sensation. Indeed, pro-inflammatory cytokines affect appetite regulation inducing loss of appetite and promote muscle catabolism [11,30,32] which have been previously reported in both renal failure [29,30] patients and animal studies [33,34]. Interestingly, in the present study, there was no measurable difference in appetite of patients with mild inflammation compared to those without inflammation. This could imply a certain threshold of inflammation that has to be exceeded before appetite is compromised. However, the size of the study population does not allow for determining the real magnitude of this threshold. The regulative role of inflammation on appetite and nutrition control has been also demonstrated in previous animal studies. Appetite behavior, nutrition sensing and energy metabolism have been shown to be under hypothalamic control. Hypothalamus receives signals about whole-body nutritional state via afferent signals from enteric nervous systems. Recent studies suggest that inflammation—occurring either within enteric and brain tissue or systemically, plays an essential role in hypothalamic adaptation [35,36].

In addition, we found a significant positive association of nutritional status and appetite scores, since the malnourished older patients revealed a worsened appetite compared to those with normal nutritional status. It should be noted that in this study, we investigated very old and critically ill patients who were admitted to a geriatric intensive care unit. Older patients often have multiple concurrent comorbidities [37,38] that may reduce appetite and food intake which results in progression of malnutrition in these patients [31].

Malnutrition is a predominant clinical risk factor with advancing age and may coexist with inflammation due to inflammaging [39] and many underlying chronic and acute diseases. Severe disease associates with tissue inflammation that results in several inflammatory responses such as anorexia and fatigue, which may suppress appetite and food intake and cause negative energy balance and development of malnutrition [31]. Nevertheless, the interplay between acute diseases, inflammation and malnutrition, all prevalent among older adults, may cause great changes on health outcomes [31,40]. The findings of our study confirm the important role of appetite as a probable key mediator between inflammation and malnutrition in older patients.

Another important finding of our study was the significant associations between self-reported appetite and measured food intake. Indeed, in the current study, almost three-quarter of subjects with very poor and poor appetite had food intake <75% of the meals offered. A good appetite is necessary to maintain sufficient food and nutrient intake and to prevent the risk of weight loss, and nutritional deficiencies [26,41] as a result poor appetite has been suggested to be one major cause of malnutrition [37]. Therefore, early determination of poor appetite and inadequate food intake in older individuals is an important issue to address, because it may allow developing effective interventions that could be performed before the onset of malnutrition and to avoid adverse health outcomes [42]. Our data suggest that appetite can closely reflect food intake in hospitalized older persons. Therefore, the measurement of food intake may be dispensable if diminished appetite is recognized in these patients.

Some limitations to the present study should be discussed. CRP alone has been used to measure inflammation whereas other inflammatory markers, which may indicate different or even stronger impact on appetite and food intake, were not considered. In addition, there are many reasons that could be responsible for diminished appetite and low food intake; therefore, residual, uncontrolled confounding cannot be excluded. Finally, it is impossible to prove causality with a cross sectional study, hence, future longitudinal research is needed to differentiate acute and chronic inflammation and the origin of inflammation to confirm causality.

## 5. Conclusions

Our findings confirm that poor appetite and low food intake are associated with inflammation in older hospitalized patients, suggesting that appetite may contribute to an important aspect in clinical health outcomes among these patients. Moreover, our data highlight that appetite is a reliable indicator in reflecting food intake in older hospitalized patients.

## Figures and Tables

**Figure 1 nutrients-11-01986-f001:**
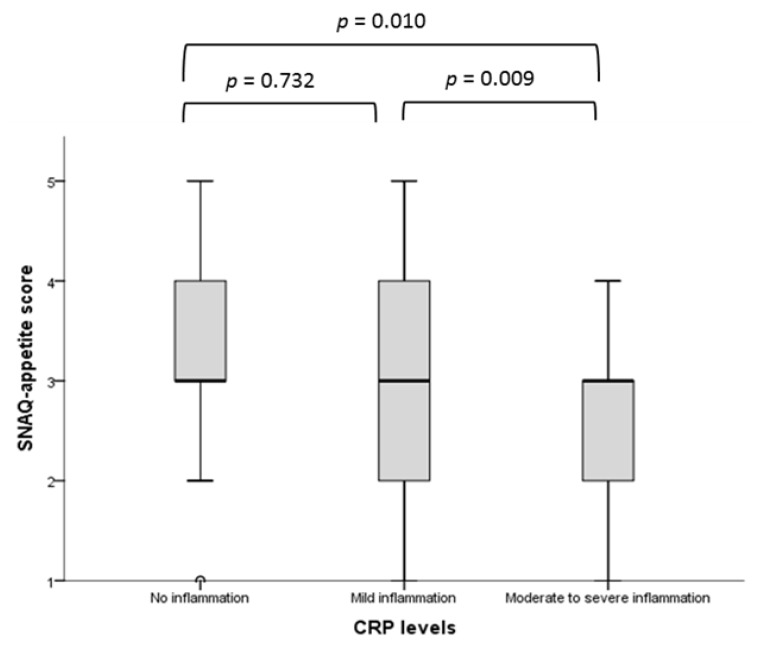
Comparison of SNAQ-appetite score across the CRP levels in total population (*n* = 200). SNAQ, Simplified Nutritional Appetite Questionnaire; CRP, C-reactive protein.

**Table 1 nutrients-11-01986-t001:** Characteristics of the study population.

	Total Population (*n* = 200)
Gender	
Female (*n*; %)	125 (62.5)
Male (*n*; %)	75 (37.5)
Age (year)	81.4 ± 6.6
Height (m)	1.66 ± 0.08
Actual body weight (kg)	73.2 ± 17.9
BMI (kg/m^2^)	26.5 ± 6.5
CRP (mg/dL)	3.5 ± 5.7
No inflammation (0–0.499 (mg/dL), *n*; %)	51 (25.5)
Mild inflammation (0.5–3.0 (mg/dL), *n*; %)	88 (44.0)
Moderate to severe inflammation (>3 (mg/dL), *n*; %)	61 (30.5)
SNAQ Score, Median (IQR)	14 (12–15)
<14 (*n*; %)	87 (43.5)
≥14 (*n*; %)	113 (56.5)
SNAQ-Appetite, Median (IQR)	3 (2–4)
Very poor and poor (*n*; %)	65 (32.5)
Average (*n*; %)	72 (36.0)
Very good and good (*n*; %)	63 (31.5)
ESAS-Appetite, Median (IQR)	2 (0–6)
No loss of appetite (*n*; %)	112 (56.0)
Average loss of appetite (*n*; %)	41 (20.5)
Severe loss of appetite (*n*; %)	47 (23.5)
MNA-SF, Median (IQR)	9 (7–10)
Normal nutritional status (*n*; %)	17 (9.0)
At risk of malnutrition (*n*; %)	119 (60.0)
Malnourished (*n*; %)	61 (31.0)
Food intake, Median (IQR)	75 (50–83)
<75% of intake	95 (48.0)
≥75% of intake	102 (52.0)
Barthel-Index on admission, Median (IQR)	45 (35–55)
Frail Simple scale score, Median (IQR)	4 (3–4)
SARC-F scores, Median (IQR)	6 (4–8)
Cognitive function (MOCA), Median (IQR)	20 (16–23)
Depression score (DIA-S), Median (IQR)	3 (1–5)
Charlson Comorbidity Index, Median (IQR)	3 (2–4)
Infection on admission	
Yes	31 (15.5)
No	169 (84.5)
Chronic inflammatory diseases	
Yes	21 (11.0)
No	166 (89.0)

BMI, body mass index; CRP, C-reactive protein; SNAQ score, Simplified Nutritional Appetite Questionnaire (maximum score 20, score <14 indicates risk of at least 5% weight loss within six months); SNAQ-appetite rated from very poor and poor (1 and 2 points), average (3 points) and good and very good (4 and 5 points); ESAS, Edmonton Symptom Assessment System; ESAS-Appetite categories as no (0–3 points), average (4–6 points) and severe (7–10 points) loss of appetite; MNA-SF, Mini Nutritional Assessment Short Form (normal nutritional status 12–14 points, at risk of malnutrition 8–11 points and malnourished 0–7 points); Food intake was measured according to the plate diagram; Frail Simple scale (not frail with score 0, pre-frail with scores of 1–2 and frail with scores of 3–5); SARC-F scores (high risk of sarcopenia with score ≥4); MOCA, Montreal Cognitive Assessment (scores <26 considered as cognitively impaired); DIA-S scores, Depression in Old Age Scale (no depressive symptom with 0–2 points, suspected depression 3 point and probable depression 4–10 points). Values are given as mean ± SD, number (%) or median (IQR, interquartile range).

**Table 2 nutrients-11-01986-t002:** Association between CRP levels and appetite scores and food intake in total population (*n* = 200).

	SNAQ-Appetite Score		ESAS-Appetite Loss Score		Food Intake (<75% or ≥75% of intake	
	Median (IQR)	*p* value	Median (IQR)	*p* value	Median (IQR)	*p* value
No inflammation	3 (3–4)		1 (0–5)		83 (58–92)	
Mild inflammation	3 (2–4)	0.003	2 (0–6)	0.013	75 (58–92)	<0.001
Moderate-Severe inflammation	3 (2–3)		5 (0–8)		58 (50–75)	

CRP, C-reactive protein; SNAQ, Simplified Nutritional Appetite Questionnaire; ESAS, Edmonton Symptom Assessment System; IQR, interquartile range; Food intake was measured according to the plate diagram.

**Table 3 nutrients-11-01986-t003:** Association between food intake and appetite scores in total population (*n* = 200).

	SNAQ-Appetite Score	ESAS-Appetite Loss Score
	*p* value	
Food intake		
<75% of intake	<0.001	<0.001
≥75% of intake		

SNAQ, Simplified Nutritional Appetite Questionnaire; ESAS, Edmonton Symptom Assessment System; Food intake was measured according to the plate diagram.

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
