# Peer review of "Inflammation, Appetite and Food Intake in Older Hospitalized Patients"

_nutrients, 2019, doi:10.3390/nu11091986_

Round 1

Reviewer 1 Report

This is a very nice study and involves an important topic in the realm of older adult malnutrition and hospital care. The following comments are provided to enhance the paper.

Abstract

1) provide cut-offs used for moderate and severe inflammation

Introduction:

2) include research questions that align with the analyses completed

Methods

3) line 99 ‘independence’ vs. independency

4) as those with cognitive impairment were excluded how does this compare to the MoCA completion and 93% of participants having impaired cognitive function? How does this impact the appetite determination that was based on patient response?

5) Comment on training of senior nurses to estimate intake. How many nurses were involved? Were they reliable in their assessment? If this is not known, it should be a noted limitation to the study.

6) What meal and admission day was used for the estimation of intake? How will this influence analyses? Was it completed on the same day as the assessment of appetite and CRP? If not controlled, then list as a limitation to the study.

7) what are the references for the cut-points for severe and moderate inflammation?

8) line 143 nutritional vs. nutritionals status

9) Also complete regression with low intake as the dependent variable to determine how inflammation predicts when adjusting for other covariates, including eating dependence, frailty, cognitive impairment and nutritional status

Results

10) line 161 infectious vs. infection diseases

11)Table 1: include CRP cut points in the table; Food intake first row should indicae median and IQR

12) The IQR on MoCA indicates this group was quite impaired; how did this influence appetite reporting?

13) Table 2: Recommend displaying CRP median and SD by SNAQ, ESAS and food intake rather than just the p values

14) line 215 predictor vs. predictors

15) quite a few statistically significant associations are ‘not shown’ and this would be helpful to see the results in tabular form

16) did the appetite ratings vary with cognitive status? Independence with eating?

Discussion

17) line 245 change relation to either relationship or association

18) line 267 ‘ who were’ admitted

19) line 282 replace ‘since’ with ‘as a result’

20) line 286 reflect vs reflects

21) limitations needs to be extended to include those noted in this review

General point

Some words like ‘thus’ ‘moreover’ and ‘furthermore’ ‘since’ are sometimes used excessively or inappropriately e.g. line 259, line 282

Reviewer 2 Report

Thank you for giving me the opportunity to review the manuscript nutrients-563910, with the title "Inflammation, appetite and food intake in older hospitalized patients". This manuscript concerns an important topic about inflammation, appetite and food intake in older hospitalized patients. The concept and addressed issue are important, but I have some points that the authors need to clarify the study designs and interpretation the outcomes.

My main point is that C-reactive protein (CPR) is an acute-phase protein, that exhibits elevated expression during inflammatory conditions such as rheumatoid arthritis, some cardiovascular diseases, and infection. The study recruited patients in a “geriatric acute care ward” and CRP was analyzed at hospital admission or the day after. The exclusion criteria did not exclude patients suffered from acute infection into analysis and the definition of “chronic inflammation disease” did not clarify in the methods. This design will make the condition of acute infection and chronic inflammatory disease become the important confounding factors to evaluate the association among CRP and other clinical scores.

The group has published a paper regarding the “The Association of Inflammation with Food Intake in Older Hospitalized Patients” in 2018【J Nutr Health Aging. 2018;22(5):589-593】. The aim of this study is to investigate the association of CRP with food intake in acutely ill older hospitalized patients. In the part of discussion, the group explore the association of CRP level and the food intake in the past 3 months prior than enrolled day. The results showed “CRP levels were not statistically significant between each groups of food intake reduction (no & mild, P=0.165; no and severe, P=0.978; mild and severe, P=0.513). The group also found that every 1mg/dl increase of CRP-level is associated with an increase in the odds of food intake <75%.  CRP levels, disease severity, mobility and BMI were the major independent predictors for food intake. I suppose the major admission reasons in this geriatric acute care ward might be acute infection, such as UTI, Aspiration pneumonia, wound or pressure sore..ect.

Therefore, from the mechanism view: CPR level reflects the active infection or disease severity (such as CKD, CAD and Rheumatoid Arthritis) of patients on admission. The activity of these disease or active infection will let patient decrease in food intake and loss appetite. Due to the current enrolled populations was in the same hospital as the previous study published in 2018 by the group, the admission criteria or the composition of the patients maybe at the same situation.

Introduction
The author's hypothesis is that the loss of appetite caused by aging is related to the inflammatory response. However, the literature review of the association between aging and CRP level is insufficiency. An important key words or term such as “malnutrition–inflammation–cachexia syndrome” and “Malnutrition-inflammation score (MIS)” did not be mentioned in the introduction.

Methods
The admission criteria of geriatric acute care ward should be mentioned. Patient with acute infection should be excluded. Otherwise, Table 1 should clarify the admission etiology of these populations.

Results
Table 1 and Table 2 should be redone as your previous study in 2018. The horizontal column should be Total populations, No inflammation, Mild inflammation and moderate to severe inflammation according to CRP level, since CRP level is your independent variable in this study.

Table 3: I don`t understand why you want to address the association between Food intake < 75% and Poor appetite and appetite loss score (SNAQ-Appetite score and ESAS-Appetite loss score). Because in this study, Food intake and appetite score were dependent variable. Actually, the correlation should be compared among CPR level and dependent variables such as Food intake, SNAQ score and ESA….

Round 2

Reviewer 1 Report

Final minor comments

Line 85- grammar

“ We also aimed to address whether patients’ appetite status reflected food intake.”

As reliability testing of three nurses was completed, this should be presented in the Methods.

Line 154 Add in ‘on food intake’

Author Response

Line 85- grammar

“ We also aimed to address whether patients’ appetite status reflected food intake.”

Thank you. It has been done (highlighted in yellow).

As reliability testing of three nurses was completed, this should be presented in the Methods.

Thank you. It has been added (lines 130-132, highlighted in yellow).

Line 154 Add in ‘on food intake’

Thank you. It has been added (line 156, highlighted in yellow).

Reviewer 2 Report

My question has been responded point by point and adequately by the author. Some important references has been added and the major questions has also been clarified. I suggest to accept their revision for publication.

Author Response

Comments and Suggestions for Authors:

My question has been responded point by point and adequately by the author. Some important references has been added and the major questions has also been clarified. I suggest to accept their revision for publication

Author:

Dear Reviewer,

Thank you so much.

Kind regards